# Redox Regulation and Glucose Metabolism in the Stallion Spermatozoa

**DOI:** 10.3390/antiox14020225

**Published:** 2025-02-17

**Authors:** Fernando J. Peña, Francisco E. Martín-Cano, Laura Becerro-Rey, Eva da Silva-Álvarez, Gemma Gaitskell-Phillips, Inés M. Aparicio, María C. Gil, Cristina Ortega-Ferrusola

**Affiliations:** Laboratory of Equine Reproduction and Equine Spermatology, Veterinary Teaching Hospital, University of Extremadura, 10003 Cáceres, Spain; femartincano@unex.es (F.E.M.-C.); lbecerror@unex.es (L.B.-R.); edasilvaz@unex.es (E.d.S.-Á.); ggaitskell@unex.es (G.G.-P.); imad@unex.es (I.M.A.); crgil@unex.es (M.C.G.); cristinaof@unex.es (C.O.-F.)

**Keywords:** stallion, spermatozoa, aerobic glycolysis, ROS, metabolism

## Abstract

Stallion spermatozoa are cells which exhibit intense metabolic activity, where oxidative phosphorylation in the mitochondria is the primary ATP generator. However, metabolism must be viewed as a highly interconnected network of oxidation–reduction reactions that generate the energy necessary for life. An unavoidable side effect of metabolism is the generation of reactive oxygen species, leading to the evolution of sophisticated mechanisms to maintain redox homeostasis. In this paper, we provide an updated overview of glucose metabolism in stallion spermatozoa, highlighting recent evidence on the role of aerobic glycolysis in these cells, and the existence of an intracellular lactate shuttle that may help to explain the particular metabolism of the stallion spermatozoa in the context of their redox regulation.

## 1. Introduction

The stallion spermatozoa are cells metabolically very active; proteomic studies provide a huge amount of data that, after thorough analysis, reveal the prevalence of pathways in the male gamete involved in metabolism and redox regulation, with numerous proteins with metabolic functions present in the sperm proteome in humans and stallions [1,2,3,4,5,6,7]. Energetic metabolism oxidizes nutrients, generating ATP; it is a series of oxidation–reduction reactions in which sugars, amino acids, and fatty acids are oxidized to simpler molecules, and the energy released in these thermodynamically favorable reactions is harnessed to phosphorylate ADP to ATP [8,9]. The tight regulation of redox reactions is essential in the metabolism. Metabolism implies the transfer of electrons from reduced organic molecules to acceptors like NAD^+^, NADP^+^ FAD, or oxygen. Reactive oxygen species like O_2_^•−^ and H_2_O_2_ are unavoidably formed, and to maintain these potentially damaging molecules under control, spermatozoa are provided with sophisticated antioxidant systems in the seminal plasma [10] and the spermatozoa themselves [11,12,13,14,15,16,17,18]. Thanks to the fundamental role of redox reactions in metabolism, metabolic activity can be estimated by measuring NADH and FAD.

This review will focus on glucose metabolism as a result of two major considerations: commercial extenders for stallion semen are formulated with high concentrations of glucose, well above physiological concentrations. A second factor is that the stallion spermatozoa could be a suitable cellular and molecular model to study glucose toxicity, with important implications for the study of diabetic conditions, so prevalent in the Western world.

Glycolysis provides two moles of ATP per mol of glucose. The glyceraldehyde 3-phosphate dehydrogenase (GA3PDH) uses NAD^+^ as an electron acceptor, generating two moles of NADH per mol of glucose. The NADH reduces pyruvate to lactate, donating electrons, and is also an important cofactor for lipid and glutathione synthesis [19]. Pyruvate enters the mitochondria, and in the mitochondrial matrix, pyruvate is oxidized and generates acetyl coenzyme A, which combines with oxalacetate to generate isocitrate and then citrate and other carboxylic acids such as ketoglutarate, succinate, and fumarate. These reactions also generate NADH and then oxaloacetate again. The succinate dehydrogenase (SDH) in the TCA cycle feeds the electron transport chain (ETC) in the mitochondria. The SDH oxidizes succinate to fumarate and at the same time reduces FAD to FADH_2_. The SDH simultaneously reduces ubiquinone to ubiquinol, with FADH_2_ oxidized to FAD. Two H^+^ and two e^−^ cross the inner mitochondrial membrane (IMM). In complex I of the ETC NADH is oxidized, releasing electrons that move along the ETC complexes; as indicated, during this process, NADH is reduced to NAD^+^. Complexes I to III pump the H^+^ derived from NADH across the IMM, and complex IV transfers e- to O_2_, the terminal electron acceptor forming H_2_O. The ATP synthase phosphorylates ADP, forming ATP using the chemiosmotic gradient generated.

Growing scientific consensus exists supporting the fact that oxidative phosphorylation is the main source of energy for motility and maintaining physically and functionally intact spermatozoa [20,21,22,23,24,25,26,27] in the stallion. However, in horses, glycolysis supports sperm velocity through glycolytic enzymes in the flagellum [25]; interestingly, while inhibition of glycolysis using 2-deoxyglucose reduces motility and velocities, it improves membrane integrity and mitochondrial membrane potential but causes a reduction in GSH content after long incubation periods [28]. Moreover, the latest evidence shows that spermatozoa have an important metabolic plasticity, being able to metabolize amino acids, sugars, and fatty acids [3], and the complex metabolism of the spermatozoa is highlighted by the discovery of the expression of the insulin receptor in them [29,30,31]. In humans, excess glucose seen in diabetic conditions is associated with infertility, and the mechanisms damaging sperm in diabetic patients share many of the aspects seen in stallion spermatozoa stored in high-glucose extenders [32,33,34,35,36,37,38,39,40,41].

It is interesting to note the scientific community’s renewed interest in studying spermatozoa’s metabolism, manifested by several recent reviews [42,43,44,45,46,47]. The present review aims to provide an updated landscape regarding the cross-link between metabolism, particularly glucose metabolism and redox homeostasis, controlling sperm survival in the equine spermatozoa.

## 2. Redox State Governs the Metabolism of the Stallion Spermatozoa

Electron transfer reactions drive all aspects of cellular metabolism. Bioenergetic metabolism is based on transferring electrons to the electron transport chain (ETC) in the mitochondria to synthesize ATP by oxidative phosphorylation [48]. Reactive oxygen, nitrogen, and sulfur species play important physiological roles, for example, in post-translational protein modifications, an aspect of utmost importance in spermatozoa due to their mostly silent DNA. However, if these compounds are deregulated, they can damage all major cellular components, proteins, lipids, and DNA. Thus, maintaining redox homeostasis is essential for all aspects of sperm metabolism, which is critical in the stallion spermatozoa. Due to the intense mitochondrial activity in this species [22,49], the stallion spermatozoa are a good cellular model for studying redox/energetic metabolism interactions (Figure 1).

All processes are tied to the cellular redox environment [48]. The redox state reflects the balance between oxidized and reduced molecules, among them, the redox pairs NAD^+^/NADH, NADP^+^/NADPH, and GSSG/GSH. In particular, the NAD^+^/NADH pair plays an essential role in the energetic metabolism of the stallion spermatozoa, with important implications for different aspects of sperm biology. These redox pairs are receiving increasing interest in many fields in biology and medicine [50,51,52] and are reviewed here in the context of the stallion spermatozoa. Spermatozoa are considered excellent cells for metabolic studies. Although these specialized cells do not grow or multiply, their internal biochemistry is highly active and they may survive for long periods both within and outside the body. They depend on extracellular sources of energy. Interestingly, the role of NAD^+^ in sperm metabolism has been well known for many decades [53,54], and even more remarkably, pioneering studies on sperm metabolism and redox homeostasis were conducted by the same group, underlining the intense relationship between them [54,55,56,57]. Moreover, spermatozoa were the first cells in which the production of hydrogen peroxide as a result of their metabolism was found [58].

## 3. Glycolysis

This is the process of the generation of pyruvate from glucose and is considered the principal pathway to obtain ATP in the spermatozoa of humans [59] and mice. In other species such as boars it is considered a significant source [60], while in stallions, ATP is mainly produced by oxidative phosphorylation in the mitochondria. However, metabolism must be viewed as an interconnected network, with important backup mechanisms that may drive metabolism depending on specific demands or physiological states.

## 4. Pentose Phosphate Pathway (PPP)

Although NADPH is also produced by the degradation of TCA cycle products, fatty acid oxidation, and ketone body utilization, the pentose phosphate pathway (PPP) is its primary source [61,62,63,64]. The PPP has two branches: an oxidative branch producing NADPH and ribonucleotides, and a non-oxidative branch with reversible reactions that interconvert glycolytic intermediates (such as fructose 6-phosphate and glyceraldehyde 3-phosphate) and pentose phosphates [62]. In the oxidative branch, glucose-6-phosphate dehydrogenase catalyzes the first reaction, dehydrogenating glucose-6-phosphate to produce NADPH and 6-phosphogluconolactone. This lactone is then hydrolyzed to 6-phosphogluconate by phosphogluconolactonase. Subsequently, 6-phosphogluconate dehydrogenase catalyzes the oxidative decarboxylation of 6-phosphogluconate, yielding another molecule of NADPH and ribulose 5-phosphate, which is then converted to ribose 5-phosphate [62]. In the context of this review, the production of NADPH to reduce oxidized glutathione (GSSG) to GSH is the main function of this pathway in spermatozoa [65,66,67,68]. Recent proteomics studies highlight the need to consider the roles of NADPH in fatty acid synthesis in spermatozoa [2], suggesting that this pathway may be active in spermatozoa [3]. After ejaculation, maintenance of redox homeostasis is considered the main function of the PPP. The PPP is therefore regulated to accelerate the oxidative branch. The non-oxidative branch is redirected to resynthesize fructose 6-phosphate, which is then converted back to glucose 6-phosphate to fuel the oxidative branch [62].

## 5. Evidence of Aerobic Glycolysis in the Stallion Spermatozoa

Sperm incorporates exogenous hexoses through specific transporters called GLUTs [69]. Within the sperm cytoplasm, glucose is phosphorylated to glucose 6-phosphate, which can then enter several pathways: the pentose phosphate pathway, glycogen synthesis, or glycolysis, producing pyruvate. Pyruvate is typically oxidized to acetyl-CoA by pyruvate dehydrogenase (PDH), reducing NAD^+^ to NADH. Although pyruvate was long considered the primary glycolytic product entering mitochondria to fuel the tricarboxylic acid (TCA) cycle, spermatozoa also convert pyruvate to lactate even under aerobic conditions. Subsequent intramitochondrial oxidation of this lactate back to pyruvate within the mitochondrial lactate oxidation complex makes lactate-derived pyruvate a key substrate for mitochondrial energetics [70]. Cellular or intracellular lactate shuttles may be of special importance in the spermatozoa. Pieces of evidence of the significance of lactate in the energetic metabolism of stallion spermatozoa have been recently reported [23], with lactate dehydrogenase detected in the mitochondrial matrix that converts lactate to pyruvate [5] and lactate seeming more efficient than pyruvate in sustaining stallion sperm motility [23], and lactate as the sole energy source has also be shown to sustain capacitation [71,72]. The importance of lactate for sperm metabolism is well known. Even in the late 1970s, Storey and Kayne [73] described the aerobic oxidation of lactate in rabbit sperm mitochondria. Monocarboxylate transporters (MCTs) have been detected in spermatozoa [70], particularly MCT1, which has been identified in the sperm head [74]. In addition, in bovine spermatozoa, lactate maintains sperm motility as well as, or better than, glucose [75]. Interestingly, in the testis, Sertoli cells secrete lactate instead of glucose to fuel sperm motility, thus the relation of Sertoli cells with spermatozoa constitutes a well-known cell-to-cell lactate shuttle [70,76].

In stallion and boar spermatozoa, oxidation of lactate to pyruvate has been observed. This oxidation, which is inhibited by the MCT inhibitor α-cyano-4-hydroxicinnamate and the LDH inhibitor oxamate, indicates that lactate is transported into sperm mitochondria for oxidation to pyruvate.

We recently described the subcellular location of three different isoforms of LDH in the stallion spermatozoa, LDHA, LDHB, and LDC, in the acrosomal region, mid-piece, and flagellum, respectively. The compartmentalization of these isoforms strongly suggests the existence of an intracellular lactate shuttle in mature spermatozoa. Moreover, functional experiments using a specific inhibitor of the isoform C (LDHC) revealed that this isoform is essential for sperm function. While the inhibition of other isoforms caused only mild, although significant, effects, a specific inhibitor of LDHC used at the same concentrations induced a rapid demise of sperm. This effect underlines the known role of the glycolytic enzymes in the sperm flagellum [77,78,79] as major contributors to energy to support motility. The activity of these enzymes was also shown in recent research from our laboratory [28]; inhibition of glycolysis with the non-metabolizable glucose analog 2-deoxyglucose (2-DG) reduced sperm motility, especially affecting sperm velocities, suggesting that the main role of these glycolytic enzymes is to provide support to motility and in particular facilitate high velocities. While the effect of 2-DG on motility may be explained by the futile phosphorylation of 2-DG reducing ATP, the greater impact on velocities is difficult to explain solely through this mechanism.

## 6. Is There an Intracellular Lactate Shuttle in the Stallion Spermatozoa?

While the presence of a cell-to-cell lactate shuttle is well described in Sertoli cells [80], the existence of a similar mechanism in the mature spermatozoa is not yet evident. In addition, stallion spermatozoa are highly compartmentalized cells with distinct roles: the head represents the cargo, paternal DNA, the mitochondria are the energy provider, and the flagellum is the propulsion system. The high compartmentalization of these cells suggests that different subcellular compartments may have different redox environments [81,82]. Although this redox compartmentalization must still be understood in the stallion spermatozoa, the use of specific fluorescent probes may indicate the existence of such redox compartmentalization [83]. The subcellular location of the three LDH isoenzymes in the spermatozoa strongly suggests the existence of an intracellular lactate shuttle in the stallion spermatozoa.

Another aspect supporting the existence of an intracellular lactate shuttle in stallion spermatozoa is the significant impact of LDHC inhibition on sperm functionality. Specific inhibition of the LDHC isoform with ethylaminooxoacetic acid (EAA) dramatically reduced motility, viability, and mitochondrial function. LDHC, the most abundant LDH isoform in sperm, and LDHA have a high affinity for pyruvate, favoring its conversion to lactate and regenerating NAD^+^ in the cytoplasm. The regenerated NAD^+^ sustains glycolysis, while lactate may enter the mitochondria, where it is converted back to pyruvate by LDHB to generate NADH. This proposed intracellular/cytosol–mitochondrion lactate shuttle [70,84,85] supports glycolysis in the cytoplasm and fuels the TCA cycle in the mitochondrial matrix, maintaining energy and redox homeostasis (Figure 2). The ultrastructural localization of these enzymes in stallion spermatozoa remains unknown and requires further research.

## 7. Dysregulated Glycolysis Causes Oxidative/Electrophilic Stress

The conversion of glucose to ATP is not a perfectly efficient process. Within the Emden–Meyerhof–Parnas pathway, phosphate groups are removed from certain intermediate metabolites, including the triose phosphates glyceraldehyde 3-phosphate (GA3P) and dihydroxyacetone phosphate (DHAP) [86]. More recently, a new enzyme has been discovered in the mammalian spermatozoa, glycerol 3-phosphate phosphatase (G3PP) [87]. This enzyme participates in the so-called “glycerol shunt”. This pathway transforms glycolysis-derived dihydroxyacetone phosphate (DHAP) to glycerol 3P and then to glycerol, and this pathway controls the levels of reactive oxygen species [87,88,89]. This new finding underlines the importance of a high regulation of metabolism to prevent oxidative stress, and this may be especially important in a cell as metabolically active as the stallion spermatozoon.

Phosphate removal from DHAP and GA3P results in the continuous production of glyoxal (G) and methylglyoxal (MG), which are 2-oxo aldehydes. These products are also generated during lipid metabolism. We recently demonstrated their formation during stallion spermatozoa storage in commercial extenders formulated with high glucose concentrations [28]. We also demonstrated that using extenders formulated with 1 mM glucose and 10 mM pyruvate maintains sperm functionality while significantly reducing the production of 2-oxo aldehydes. The adjacent carbonyl groups of these compounds make them strong electrophiles that readily react with nucleophiles present in proteins, lipids, and DNA, resulting in the formation of advanced glycation end products (AGEs) [90]. This is a similar landscape as occurs with hydrogen peroxide (H_2_O_2_) and/or the superoxide anion O_2_^•−^, which act as regulatory molecules but may trigger significant cellular damage if redox homeostasis is lost. Glutathione (GSH) conjugation serves as a control mechanism for 2-oxoaldehydes [9]. The recent discovery of the SLC7A11 x-CT glutamate/cystine antiporter in stallion spermatozoa highlights the importance of glutathione for sperm functionality [91,92]. This antiporter, which exchanges intracellular glutamate for extracellular cystine, is constitutively expressed in spermatozoa. This constitutive expression is rare, occurring in only a few cell types, including those of the thymus, spleen, and brain [93]. mRNA has been detected in the testis, and the SLC7A11 knock-out mouse displays subfertility [94]. This protein is upregulated in many cancer cell lines [95,96]. Once incorporated, cystine is reduced intracellularly to cysteine and used for GSH synthesis [92,97]. Evidence for GSH synthesis in spermatozoa includes the identification of the necessary enzymatic machinery—glutathione synthetase (GSS) and gamma-glutamylcysteine ligase (GCLC)—and functional studies using the GCLC-specific inhibitor L-buthionine sulfoximide (BSO) combined with GSH measurement via mass spectrometry [97]. Although spermatozoa are generally considered translationally silent, recent claims suggest they may actively synthesize the enzymes involved in GSH synthesis [98]. The possibility of limited translation of specific proteins warrants further investigation in light of new evidence.

## 8. Excess Glucose Predisposes Stallion Spermatozoa to Ferroptosis

Excess glucose not only leads to redox deregulation and oxidative stress but also predisposes the stallion spermatozoa to ferroptosis [91,99] (Figure 3). Commercially available sperm storage media expose spermatozoa to glucose concentrations of 67 mM or even higher, far exceeding physiological levels. While equine serum glucose is around 5 mM, oviductal glucose concentrations are much lower, peaking at approximately 300 μM [100]. Obviously, extenders in use expose stallion spermatozoa to supraphysiological glucose concentrations, potentially leading to glucose toxicity [101], involving different mechanisms. This process encompasses several key events, including the production of oxoaldehydes (detailed in the previous section), the direct induction of reactive oxygen species (ROS) by glucose, the activation of MAP kinase signaling pathways, mitochondrial fission mediated by calcium ions (Ca^2+^) [102,103], and activation of the polyol pathway which consumes NADPH leading to GSH depletion and this pathway which also consumes NAD^+^, compromising glycolysis [104]. Hyperglycemia activates a distinct metabolic route characterized by the involvement of diacylglycerol (DAG), protein kinase C (PKC), and NADPH-oxidase. This activation results in the overproduction of reactive oxygen species (ROS) and subsequent mitochondrial damage. The resulting mitochondrial dysfunction may inhibit glyceraldehyde 3-phosphate dehydrogenase (GAPDH), leading to the diversion of glycolytic metabolites upstream of this enzyme. This diversion increases the flux of dihydroxyacetone phosphate (DHAP) towards diacylglycerol, thereby further activating protein kinase C (PKC) and perpetuating the cycle [103,104]. Furthermore, DHAP is the precursor of the oxoaldehyde MG [105,106]. In addition, high glucose concentration predisposes cells to apoptosis, ferroptosis, necroptosis, and other types of cell death [107]. Recent investigations have demonstrated that exposure of spermatozoa to high glucose concentrations in vitro may render these cells susceptible to ferroptosis. This form of regulated cell death is characterized by its dependence on ferrous iron (Fe^2+^) and is initiated by the formation of lipid peroxides. A critical feature of ferroptosis is the compromised function of the glutathione (GSH)-dependent antioxidant defense mechanism [108]. The recent publication of evidence for ferroptosis in stallion spermatozoa represents a new finding [91], showing that the induction of ferroptosis in stallion spermatozoa is facilitated by extension in high-glucose media [99]. Excess glucose, by driving excessive oxoaldehyde production, can compromise cellular antioxidant defenses through GSH depletion [105,106]. Glyoxal (G), methylglyoxal (MG), and advanced glycation end products have been identified as inducers of ferroptosis [109].

## 9. Conclusions

Spermatozoa exhibit intense energetic demands that vary throughout their lifespan. From their origin in the germinal epithelium to their transit through the female reproductive tract and subsequent fertilization, adequate energy sources are essential for processes such as capacitation. The inherent nature of energetic metabolism, involving numerous oxidation–reduction reactions, leads to the unavoidable production of reactive oxygen species (ROS). However, precise control of ROS production is necessary for maintaining sperm functionality. Therefore, the study of the complex interactions between metabolism and redox homeostasis represents a critical area of investigation for advancing our understanding of male factor infertility and developing improved sperm biotechnologies. Particularly relevant is the effect of supraphysiological concentrations of glucose in commercial extenders for stallion spermatozoa. Stallion spermatozoa are exposed to elevated glucose concentrations in commercial extenders (above 67 mM compared with 5 mM in plasma), which induces cellular damage that mimics some aspects of diabetic conditions. A particularly relevant mechanism is the glucose-induced production of reactive oxygen species (ROS). This oxidative stress contributes to decreased sperm motility, viability, and mitochondrial function, mirroring the detrimental effects of hyperglycemia observed in diabetic patients. Furthermore, our data suggest that glucose-induced ferroptosis may also play a role in this damage [99].

## Figures and Tables

**Figure 1 antioxidants-14-00225-f001:**
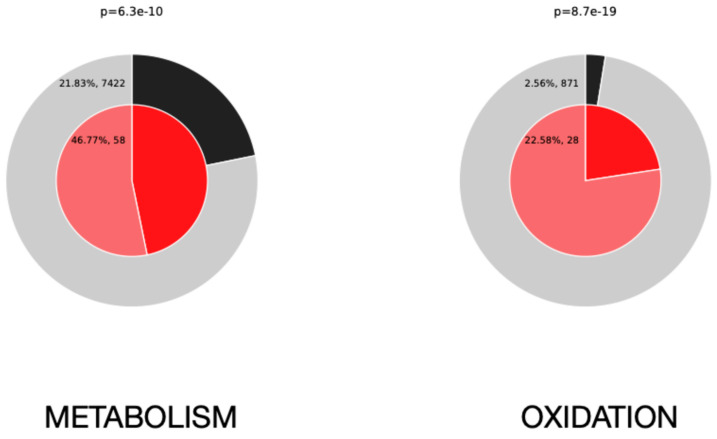
Importance of the metabolism and redox reactions in the stallion spermatozoa. Enrichment of proteins matching membership terms: “metabolism” on the left and “oxidation” on the right. The outer and inner pie charts show the proportion of proteins within the background (whole proteome, outer, in black) and input (sperm proteome, inner in red) datasets, respectively. The *p*-values indicate that both memberships (metabolism and oxidation) are statistically significantly enriched on the lists. https://metascape.org/gp/index.html#/main/step1 (accessed on 27 July 2024).

**Figure 2 antioxidants-14-00225-f002:**
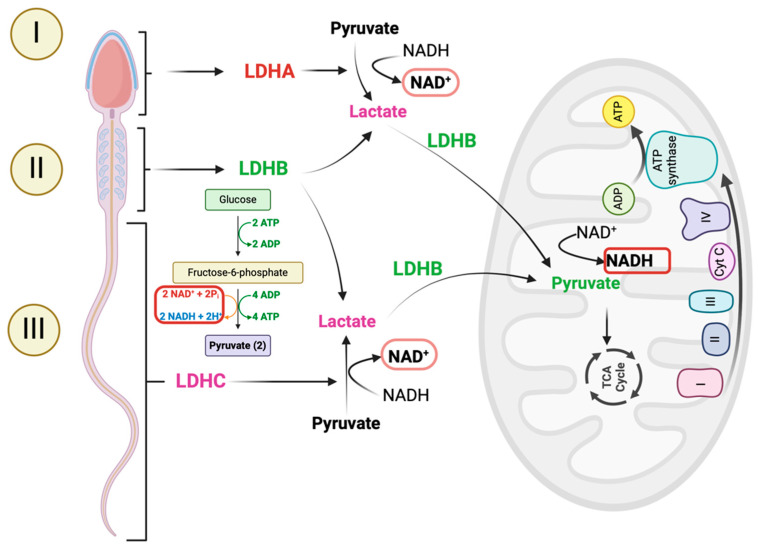
Proposed mechanism of the intracellular lactate shuttle in the stallion spermatozoa. Isoforms of LDH are highly compartmentalized in the spermatozoa. LDHA and LDHC expressed in the acrosomal region (I) and tail (III), respectively, have a higher affinity for pyruvate; these isoenzymes reduce pyruvate to lactate, and the electron donor is NADH, which is oxidized to NAD^+^. NAD^+^ favors glycolysis, supporting glycolytic enzymes in the flagellum (III) and improving sperm velocity. The lactate generated is imported into the mitochondria, supporting mitochondrial function; in the mitochondria, the isoform present is LDHB (II) with a higher affinity for lactate. This is oxidized to pyruvate which feeds the TCA cycle, providing reducing equivalents and donating electrons to the ETC to generate energy to phosphorylate ADP, producing ATP.

**Figure 3 antioxidants-14-00225-f003:**
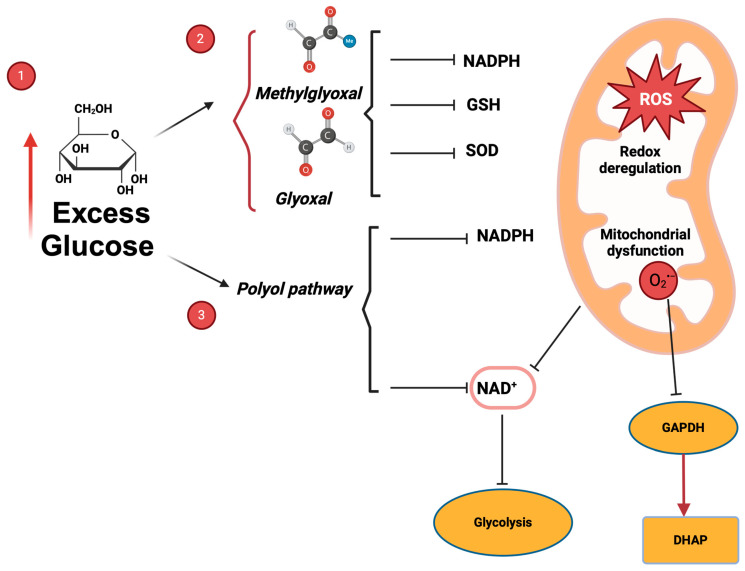
(1) Excess glucose in the media triggers several damaging pathways. (2) During glycolysis, the removal of phosphate groups from glyceraldehyde 3-phosphate (GA3P) and dihydroxyacetone phosphate (DHAP) leads to the formation of reactive oxoaldehydes, such as glyoxal and methylglyoxal. These compounds, due to their highly reactive carbonyl groups, act as strong electrophiles, capable of oxidizing lipids, proteins, and DNA. Their detoxification relies on the glutathione (GSH)-dependent glyoxalase system, which consumes both GSH and NADPH. Furthermore, glyoxal can directly deactivate superoxide dismutase (SOD), further compromising antioxidant defenses. (3) Excess glucose can also be shunted into the polyol pathway, where it is reduced to sorbitol and then oxidized to fructose. These reactions consume NADPH, disrupting redox homeostasis, and NAD^+^, impacting glycolysis. Finally, mitochondrial dysfunction contributes by increasing superoxide (O_2_^•−^ production, which inhibits glyceraldehyde 3-phosphate dehydrogenase (GAPDH), further disrupting glycolysis and leading to an accumulation of oxoaldehyde precursors, thereby exacerbating oxoaldehyde formation.

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
