# Peer review of "Redox Regulation and Glucose Metabolism in the Stallion Spermatozoa"

_antioxidants, 2025, doi:10.3390/antiox14020225_

Round 1

Reviewer 1 Report

Lines 19, 39 qnd 76 : please clarify the main focus of the review uniquely

Lines 78, 79:, "controlling sperm survival...." change in "controlling stallion sperm survival"

Line 121 add In other species

Lines 267-269 please clarify this sentence:what are the physiological levels?

No comments

Author Response

Lines 19, 39 qnd 76 : please clarify the main focus of the review uniquely

Done

Lines 78, 79:, "controlling sperm survival...." change in "controlling stallion sperm survival"

We appreciate the commentary, but since the sentence  ends “….in the equine spermatozoa, adding 'stallion' may be redundant

Line 121 add In other species

Done

Lines 267-269 please clarify this sentence:what are the physiological levels?

This is already indicated at the beginning of section 8, lines 271-272 in the revised version, 5mM in serum, 300 micromolar in the oviduct

Reviewer 2 Report

I genuinely enjoyed reading this interesting review which highlights interesting topics in sperm physiology, and an interesting perspective in their application as a model for glucose metabolism in correlation with diabetes. However, I missed the discussion regarding the last part, that is how all the data and information provided in the eight different sections of the manuscript correlate to the application of stallion spermatozoa in the study of diabetic conditions.

Author Response

Reviewer 2

I genuinely enjoyed reading this interesting review which highlights interesting topics in sperm physiology, and an interesting perspective in their application as a model for glucose metabolism in correlation with diabetes. However, I missed the discussion regarding the last part, that is how all the data and information provided in the eight different sections of the manuscript correlate to the application of stallion spermatozoa in the study of diabetic conditions.

We appreciate the commentary, effectively, this is an interesting aspect that is now included in the discussion of the revised version

Reviewer 3 Report

Lines 45-61 are not entirely correct: Indeed glycolysis of one molecule glucose is resulting in the net formation of 2 molecules pyruvate 2 molecules of ATP and 2 molecules of NADH. These two NADH molecules formed are used for fermenting the formed pyruvate into lactate. Thus the net production of glycolysis followed by lactate formation is :

1 glucose + 2ADP --> 2 lactate + 2 ATP (no net formation of NADH).

Moreover for  lipid synthesis not NADH but instead NADPH is an important cofactor (for fatty acid synthesis) likewise reduction of glutathione involves NADPH instead of NADH. NADH is also involved but only at a later step in lipid synthesis as it is required for converting dihydroxyacetone phosphate into glycerol 3-phosphate.

Line 127 beyond the oxidative steps in the pentose phosphate pathway the most important other NADPH forming step is the combined dehydrogenation and decarboxylation of malate into pyruvate by malic enzyme. 

Line 151-155 and later in that paragraph:  The presence of the lactate production under aerobic conditions doe not necessarily come from an intramitochondrial source. Proper studies on the presence of LDH distribution in the low amount of cytosol and in the matrix of the mitochondria should show specific isoforms for that claim. Perhaps the amount and distribution of LDH may give a clue on species specificity on dependence of aerobic versus anaerobic functioning of sperm.

Figure 2 and Line 175: is the LDHB indeed present in the matrix of mitochondri?  has that been confirmed by transmission electron microscopy (TEM) of ultrathin sections of  immunogold labelled mid-piece?

Likewise LDHA is that in the acrosome (likely not)? or is it in the area between the acrosome and plasma membrane as evidenced transmission electron microscopy (TEM) of ultrathin sections of  immunogold labelled sperm head?

Likewise LDHC is that in the principle piece in the fibrous sheath and or the outer dense fibers? as evidenced by transmission electron microscopy (TEM) of ultrathin sections of  immunogold labelled principal piece.

I think such ultrastructural information is important and if not available from the literature the authors alt least must address that the ultrastructural localization of these isoforms of LDH must be studied.

No further details

Author Response

Lines 45-61 are not entirely correct: Indeed glycolysis of one molecule glucose is resulting in the net formation of 2 molecules pyruvate 2 molecules of ATP and 2 molecules of NADH. These two NADH molecules formed are used for fermenting the formed pyruvate into lactate. Thus the net production of glycolysis followed by lactate formation is :

1 glucose + 2ADP --> 2 lactate + 2 ATP (no net formation of NADH).

Moreover for  lipid synthesis not NADH but instead NADPH is an important cofactor (for fatty acid synthesis). likewise reduction of glutathione involves NADPH instead of NADH. NADH is also involved but only at a later step in lipid synthesis as it is required for converting dihydroxyacetone phosphate into glycerol 3-phosphate.

Effectively, the reviewer is right, however, the aim of the paragraph is to focus on the NADH/NAD+ donating and accepting electrons in the metabolism, not the metabolism itself. We indicate that NADH is a cofactor in lipid synthesis for the same reason.

Line 127 beyond the oxidative steps in the pentose phosphate pathway the most important other NADPH forming step is the combined dehydrogenation and decarboxylation of malate into pyruvate by malic enzyme. 

Yes, the reviewer is right, and we appreciate the commentary, but our review focused on glucose, and we indicated that PPP is the main source, not the unique source of NADPH

Line 151-155 and later in that paragraph:  The presence of the lactate production under aerobic conditions doe not necessarily come from an intramitochondrial source. Proper studies on the presence of LDH distribution in the low amount of cytosol and in the matrix of the mitochondria should show specific isoforms for that claim. Perhaps the amount and distribution of LDH may give a clue on species specificity on dependence of aerobic versus anaerobic functioning of sperm.

We appreciate the pertinent commentary of the reviewer, and we agree. In fact, in the next section, number 6 all these aspects are discussed

Figure 2 and Line 175: is the LDHB indeed present in the matrix of mitochondri?  has that been confirmed by transmission electron microscopy (TEM) of ultrathin sections of  immunogold labelled mid-piece?

Likewise LDHA is that in the acrosome (likely not)? or is it in the area between the acrosome and plasma membrane as evidenced transmission electron microscopy (TEM) of ultrathin sections of  immunogold labelled sperm head?

Likewise LDHC is that in the principle piece in the fibrous sheath and or the outer dense fibers? as evidenced by transmission electron microscopy (TEM) of ultrathin sections of  immunogold labelled principal piece.

I think such ultrastructural information is important and if not available from the literature the authors alt least must address that the ultrastructural localization of these isoforms of LDH must be studied.

 I think that the reviewer is raising very interesting ideas, and a commentary on the need for further studies is indicated as suggested in the revised version lines 226-227

Reviewer 4 Report

Because stallion spermatozoa may serve as an appropriate cellular and molecular model for glucose toxicity studies, this paper focuses on stallion spermatozoa and discusses elaborate mechanisms for maintaining metabolic and redox homeostasis based on recent findings regarding glycolysis and the presence of intracellular lactic acid shuttles. The authors have actively published a number of review articles on this particular topic in recent years, and this paper will be of interest to readers as an update of those.

Since this abstract does not clearly indicate what this paper is about, it should be changed to one that explains why it is stallion sperm and why it is glucose metabolism.

Cited papers 1 and 2 of the proteomics data 1-7 are human sperm papers, so the references should be removed or reworded to avoid misunderstanding.

Why is it that “commercial diluents for stallion semen are formulated with high concentrations of glucose that far exceed physiological concentrations.” Please explain briefly how this has been done.

Please fix the places where the text should be in superscript but isn't.

For example, Ca2+ in line 275. There were also cases where there were no periods or too many periods.

Author Response

Reviewer 4

Because stallion spermatozoa may serve as an appropriate cellular and molecular model for glucose toxicity studies, this paper focuses on stallion spermatozoa and discusses elaborate mechanisms for maintaining metabolic and redox homeostasis based on recent findings regarding glycolysis and the presence of intracellular lactic acid shuttles. The authors have actively published a number of review articles on this particular topic in recent years, and this paper will be of interest to readers as an update of those.

We appreciate the commentaries of the reviewer

Since this abstract does not clearly indicate what this paper is about, it should be changed to one that explains why it is stallion sperm and why it is glucose metabolism.

Following the commentaries of the reviewer, the abstract has been modified accordingly

Cited papers 1 and 2 of the proteomics data 1-7 are human sperm papers, so the references should be removed or reworded to avoid misunderstanding.

Done as suggested

Why is it that “commercial diluents for stallion semen are formulated with high concentrations of glucose that far exceed physiological concentrations.” Please explain briefly how this has been done.

We are not sure we understand what the reviewer is trying to say, but we have clarified that most commercial extenders have glucose concentrations greater than 67 mM.

Please fix the places where the text should be in superscript but isn't.

For example, Ca2+ in line 275. There were also cases where there were no periods or too many periods.

We have revised the document accordingly

Round 2

Reviewer 3 Report

No further comments

Noi further comments